

**Hydraulic and geochemical impact of occasional saltwater intrusions**
**through a submarine spring in a karst and thermal aquifer (Balaruc**
**peninsula near Montpellier, France)**
Marie-Amélie Pétré[1,2*], Bernard Ladouche[3], Jean-Luc Seidel[1], Romain Hemelsdaël[4],
Véronique de Montety[1], Christelle Batiot-Guilhe[1], Claudine Lamotte[3]
[1]HydroSciences Montpellier, Montpellier University, CNRS, IRD, Montpellier, France
[2]Now at North Carolina State University, Marine, Earth, and Atmospheric Sciences Department,
Raleigh, NC 27695, United States
[3]BRGM, Montpellier University, Montpellier, France
[4]Geosciences Montpellier, Montpellier University, CNRS, Montpellier, France
*corresponding author: Marie-Amélie Pétré, marieamelie.petre@gmail.com
**Abstract**
Submarine springs are a common discharge feature of the karst aquifers along the
Mediterranean coast. In some instances, occasional and localized saltwater intrusions can occur
through the submarine spring and negatively impact the quality of the groundwater resource.
The hydraulic and geochemical behavior of a submarine spring discharging into the Thau
lagoon just offshore of the Balaruc peninsula near Montpellier, France has been characterized
to determine the impacts of such phenomena to better understand the dynamics of a regional
karst aquifer and improve its groundwater management. This work is based on both historical
and new hydrogeological and geochemical data, illustrating 6 occasional saltwater intrusion
events (from 1967 to 2014) in the Thau lagoon area (southern France).
Hydraulic perturbation of the aquifer is propagated instantly within the Balaruc-les-Bains
peninsula and reaches a distance of about 5 km upgradient within 9 days. Comparison of
hydraulic heads during seawater intrusion events in 2010 and 2014 indicates an aggravation of
the phenomenon with an increase in hydraulic head variations.
In contrast, isotopic tracers ($^{87}Sr/^{86}Sr$, D/H, $^{18}O/^{16}O$,) and Rare Earth Elements (REE)
demonstrate that the geochemical impact of these inversac events is only observed at the local
scale, but is still perceptible several years after the event. For example, some of the thermal
wells had not recovered their initial geochemical state 20 and 40 months after the last two
inversac events (2010 and 2014, respectively), suggesting a geochemical legacy of this
phenomenon within the complex karst system. By contrast, an adjacent deep karst compartment


located south of the study area is not affected by the saltwater intrusion and is characterized by
distinctly different hydrodynamic behavior.
Overall, this work on occasional and localized saltwater intrusions constitutes a key step in
understanding the dynamics of this complex karstic and thermal aquifer and will support the
management of the groundwater resource.

## 1. Introduction


Submarine springs are common discharge features of karst systems along the Mediterranean
coast (Bakalowicz, 2014; Fleury, 2005; Fleury et al., 2007; Stieglitz et al., 2013). Water from
these springs can support economic development or be exploited for drinking water supply.
In certain instances, especially during periods of low discharge, flow can reverse and instead
intrude into the upgradient karst aquifer, causing an adverse impact on the quality of the
groundwater resource and the economic activities that depend on it (Arfib and Gilli, 2010;
Drogue and Bidaux, 1986). This phenomenon, termed "*inversac*" in French (Gèse, 1987;
Pinault et al., 2004), was previously described by Albéric (2004) and Joigneaux et al. (2011) in
a related context of river backflooding into karst springs.
The factors responsible for initiating and terminating inversac events are not particularly well
understood and the recovery of carbonate aquifers after a seawater intrusion is poorly described
(Han et al., 2015). Yet, it is necessary to understand the mechanisms controlling this
phenomenon, as well as the long-term impact of occasional and localized saltwater intrusions,
to appropriately manage the groundwater resources of such karst systems.
In this context, a hydraulic and geochemical study was undertaken to characterize the
occasional and localized saltwater intrusions into the karst aquifer of the Thau basin. More
specifically, our study addressed the following questions: (i) What are the hydraulic and
geochemical conditions in the karstic aquifer during and after an inversac event? (ii) What are
the triggering factors contributing to the occurrence of the inversac event? (iii)To what extent
is the aquifer system able to recover from an inversac event and get back to its original
geochemical state? (iv) What are the implications in terms of groundwater resources
management?
The Thau hydrosystem in the Balaruc-les-Bains area is an example of a karst and thermal
aquifer affected by occasional saltwater intrusions through a submarine spring. This complex
hydrologic system has been described by Aquilina et al. (1997, 2002, 2003), Doerfliger et al.





(2001) and Ladouche et al. (2001). However, the causes and extent of the impact of the saltwater
intrusions here remain poorly understood. The karst and thermal water resources across the
Thau hydrosystem are exploited both for drinking water supply and spa activities. Balaruc-les-
bains (Fig.1) is the largest spa in France with 55,000 visitors annually. In the past, water from
the Vise submarine spring was diverted to the seashore and used for aquaculture activities.
Additionally, the Thau lagoon supports an extensive shellfish aquaculture and fishery. Thus,
the economic stakes are high in this area and water use conflict has the potential to arise between
stakeholders during dry conditions, as a consequence of an increase in water demand, and the
occurrence of occasional saltwater intrusions (Chu et al., 2014; La Jeunesse et al., 2015).
Indeed, the karst system has been affected by six inversac events through the Vise submarine
spring over the past 50 years (Table 1), five of which (1967,1983, 2008, 2010 and 2014) have
been documented and described (Ladouche et al., 2011, 2019; Ladouche and Lamotte, 2015).
Human activities, such as bauxite mining upstream from the Vise spring, and intervention on
the spring itself are thought to have contributed to the saltwater intrusions over the 1967-1993
period by drawing down the hydraulic head of the karst system. Historically, these inversac
events produced an increase in the water level and electrical conductivity at the observation
points (thermal well and spring) in the Balaruc peninsula. A better understanding of this
phenomenon is for local use purposes, but is also of strategic importance for water managers of
the Issanka karst spring, a major drinking water supply in the area.
The last two inversac events in 2010 and 2014 lasted seven and five months, respectively. The
2014 inversac event led to the permanent abandonment of the Cauvy spring which provided
drinking water to a local population of 30,000.









| Date | Duration | Causal Factors | End of Episode |
|---|---|---|---|
| **2014** (May-October) | 5 months | Very low waters. No storm surge in the lagoon. Groundwater withdrawals | Temporary low water level in the lagoon combined with recharge in the karst aquifer |
| **2010** (June-December) | 7 months | Intermediate waters | Heavy rainfall |
| **2008** (January-March) | 3 months | Low waters and a pump test on thermal well F14 | - |
| **1993** (October) | 24 days | Low waters. Human intervention on the submarine spring and pumping test on the Balaruc peninsula | Heavy rainfall |
| **1984-1987** (May-February) | 20 months | Low waters. Lagoon with high water level. Human intervention on the griffon of the submarine spring | Heavy rainfall |
| **1967** (October) - **1969** (March) | 17/18 months | Dry conditions. Bauxite mining at the Cambelliès site. Large pumping (up to 1000m³ h⁻¹), that lowered the water level to -20 m asl | Cessation of groundwater pumping |


Table 1. Summary of the observed occasional saltwater intrusions (inversac events) in the karst
and thermal aquifer of the Thau basin.
A multi-tracer approach was followed to better define the origin and contribution of the thermal,
karst and saltwater end-members under different conditions. Water stable isotopes (D/H,
$^{18}O/^{16}O$) were used to quantify the contribution of the saltwater end-member, whereas strontium
isotopes ($^{87}Sr/^{86}Sr$) were used to distinguish the origin of salinity (thermal or marine). Rare
Earth Elements (REE) were used to establish the chemical signature of the thermal wells in a
reference context and to evaluate the influence of the marine signature over time. Finally, the
B- isotope signature ($d^{11}B$) was used to trace water-rock interaction initiated during inversac
events.

## 2. Study Area and Geological Setting

The Balaruc peninsula is located along the Mediterranean coast of southernmost France, west
of the city of Montpellier, (Fig. 1a). The karst aquifer of the peninsula lies within the Middle
and Upper Jurassic carbonates that outcrop to the north in the Aumelas Causse and the Gardiole
Massif. These Jurassic carbonates are buried under Miocene to Plio-Quaternary deposits in the
eastern part of the Thau lagoon (Fig 1b). Karstification and burial of the main Jurassic aquifer
are the result of the geological history of the region described below.





A Mesozoic series of Triassic to Early Cretaceous age was deposited during Tethyan rifting (Baudrimont and Dubois, 1977; Debrand-Passard, 1984). The Mesozoic carbonate platform underwent uplift during the Mid-Cretaceous, leading to exhumation that caused erosion of the Early Cretaceous series and strong karstification of the Jurassic carbonates

The Mesozoic series was successively deformed during the Pyrenean collision phase from Upper Cretaceous to Eocene time (Arthaud and Laurent, 1995; Arthaud and Seguret, 1981; Choukroune et al., 1973; Choukroune and Mattauer, 1978) and the later rifting of the Gulf of Lion (Arthaud et al., 1977; Arthaud and Seguret, 1981; Benedicto et al., 1996; Maerten and Séranne, 1995; Séranne, 1999; Thaler, 1962). Then, the sea level rise of the Early Miocene (Burdigalian) deposited transgressive sediments across the region (Fig. 1b). The overall sedimentary record associated with these Upper Cretaceous to Middle Miocene geological events, includes a wide range of detrital and carbonate facies deposited in marine, lacustrine and fluvial environments (Combes, 1990; Marchand, 2019) (Fig. 1a).

The sea level fall related to the Messinian Salinity Crisis caused major river incision across southern France which, as a consequence, deepened the karst system (Clauzon, 1982; Hsu, 1973; Ryan, 1976). The Messinian paleovalley in the coastal area was then flooded during the early Pliocene when the Mediterranean was connected again to the Atlantic Ocean. Recent sea level fluctuations over the last 5 Ma and associated Plio-Quaternary deposits have generated the present day morphology of the Thau lagoon.

Structurally, the Balaruc-les-bains area is located between both the major NE-trending Cevennes and Nîmes faults, to the south of the Pyrenean Montpellier Thrust (Fig. 1a). The study area is also affected by the presence of the E-W oriented Pyrenean Thau Thrust. The complex fault network at the intersection with the Nîmes Fault allows to bring the Paleozoic basement at shallow depth (less than 2000 m according to borehole data) in the Balaruc-Sète area (Fig. 1a)





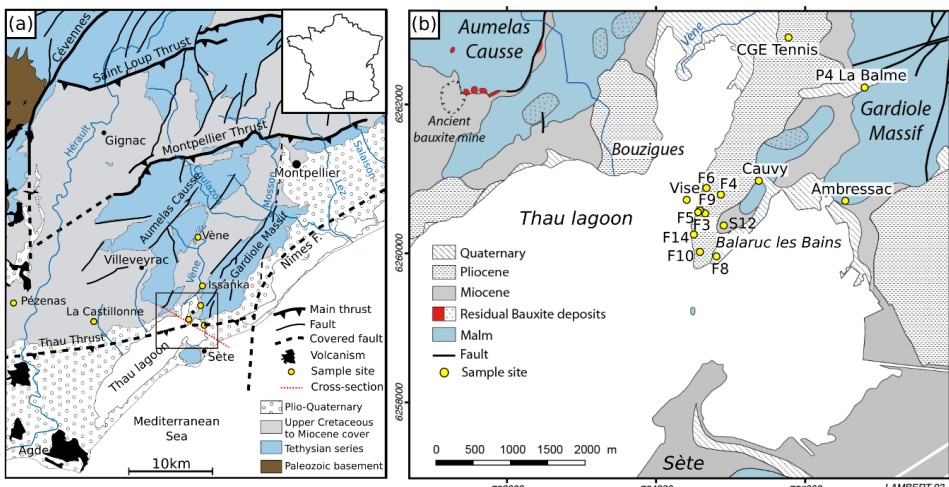

138

*Figure 1 (a) Simplified geological map showing the main structures of the Montpellier region. The box locates the*
*Balaruc-les-bains study area, between the Montpellier thrust and the Nîmes Fault. (b) Detailed geological map*
*of the eastern Thau lagoon. Sample sites indicated by the yellow dots in both maps correspond to the locations of*
*the thermal wells, karst springs, and other piezometric drilling sites. The location of the ancient bauxite mine*
*(Cambelliès) is indicated by the dashed oval in the northwest of the study area.*

## 3. Hydrogeological conceptual model of the Thau hydrosystem

The Balaruc peninsula is located at the point of natural convergence of different types of
waters including seawater from the Thau lagoon and Mediterranean Sea, cold karst
groundwater sourced from the Aumelas Causse and Gardiole Massif, and mineralized hot
thermal water rising from depth in the Balaruc peninsula (Aquilina et al, 2002). These different
types of water interact together in the Middle to Upper Jurassic carbonate reservoir. The
Montpellier Thrust to the north of the study area (Fig. 1a) brings impermeable marly Lias to
surface, which delimits the aquifer and acts as barrier to groundwater circulation. Groundwater
flow follows a general N-S trend from the Aumelas Causse and Gardiole Massif to the Balaruc
peninsula and the Mediterranean Sea. A clear hydraulic connection from the Vène spring
(Aumelas Causse) to Issanka spring (western slope of the Gardiole Massif) has been
demonstrated through artificial tracer testing (Paloc et Bonnet, 1969; Ladouche et al., 2001).
The main outlets of the Aumelas Causse karst network are the Vise submarine spring, the
Issanka spring and the Vène spring which acts as an overflow outlet. The Vise submarine spring
is located on the floor of the Thau lagoon at 30 m depth and constitutes the most downgradient
discharge feature of the hydrological system. The western part of the Gardiole Massif is drained
by the Vise spring as well as Cauvy and Ambressac springs.



At the regional scale, four levels of groundwater circulations have been recognized by previous
studies (Aquilina et al., 2003): 1) a surficial groundwater flow component originating from the
recharge areas that has a residence time <20 years; 2) an intermediate groundwater flow system
from the Aumelas Causse, that circulates below the Montbazin-Gigean basin and the Gardiole
Massif and discharges at points along the continental shelf (residence time of 50 years); 3) a
deep regional groundwater flow system directed from the Aumelas Causse to the Villeveyrac
Basin where several deep drillings (Pézenas and La Castillonne, depth of 1000 m) present hot
karstic waters with low mineralization (residence time of several thousand years); 4) a very
deep (>2 km) paleo-karst water circulation mixed with an ancient seawater circulation both
recharging the thermal reservoir that has residence times on the order of 100,000 years.

## 4.   Material and methods

### 4.1 Hydrogeological Monitoring

Water level, specific electrical conductivity and temperature were measured at 10 locations on
an hourly basis during the last three inversac events in 2008, 2010 and 2014. The time-series
data used in this study come from the "national quantitative groundwater monitoring network
of the AFB/BRGM (available on the ADES website: https://ades.eaufrance.fr for P4 La Balme
(location 10166X0253) and CGE Tennis (location 10166X0212.) Other observation stations
(F5, F6, F8, F9 and S12) are managed by Balaruc-les-Bains and the municipality of Sète and
the Syndicat Mixte du bassin de Thau (El Cantou, F5-Issanka, Frescaly and Cauvy).
In order to compare the different inversac events, hydraulic head and electrical conductivity
were expressed with respect to reference values observed before the inversac events, as
discussed in the Results Section below.  In Figures 3 and 5) b and 5d), the x-axis corresponds
to the number of hours elapsed since the beginning of the inversac event (t0). The y-axis
represents the difference between the parameter (hydraulic head or electrical conductivity)
measured during the inversac and the preceding respective reference values. The variable "delta
H" is the difference between hydraulic head measured at time ="t" - hydraulic head observed
at time= $t_0$ (before the inversac event). Thus, a positive value corresponds to an increase
compared to the baseline condition and vice versa. This relative comparison permits the
evaluation of the inversac event perturbations with respect to the initial conditions prevailing
in the system and result in a more precise valuation of the transient states of the system.



### 4.2 Geochemical and Isotopic Analysis

Geochemical data for the 1996-2000 period are from Aquilina et al. (2002, 2003) and Ladouche et al. (2001). They are considered as representative of reference geochemical conditions of the karst system. Indeed, these data were collected three to seven years after the 1993 inversac, which is considered as a sufficient time for the system to have returned to its equilibrium state given the short, 21-day interval of this event.

New geochemical data (major ions and REE compositions) and isotope analyses (D/H, $^{18}O/^{16}O$, $^3H$, $^{87}Sr/^{86}Sr$ and $d^{11}B$) were determined for samples collected in October 2010, April and September of 2012 and March and August of 2018, i.e. during the 2010 inversac, 17 and 22 months after the end of the 2010 inversac and 3.5 years after the end of the 2014 inversac.

Temperature, pH, Dissolved Oxygen and Electrical Conductivity ($T_{ref}$=25 °C) were measured in the field, using a portable pH meter, oxymeter and conductivity meter (WTW 3210).

Water sample for major and trace element analysis were filtered on-site with disposable PP syringe with a 0.22 µm Durapore membrane and stored in acid washed HDPE bottles. Aliquots for cations and trace elements were acidified to pH 2 with ultrapure $HNO_3$ (1‰ v/v). Samples for H- and O-isotope analysis were collected in 15 mL amber glass vials capped with airtight lids. One liter samples were collected for B- and Sr-isotope analysis in pre-cleaned HDPE bottles and were later filtered in a cleanroom through a 0.22 µm Durapore membrane in a pressurized Nalgene filtration unit, with samples for Sr-isotope analysis acidified with 1 % ultrapure $HNO_3$. Samples for tritium measurement were collected in 1 L HDPE bottles. All samples were stored at 4 °C before analysis. Chemical analyses were performed in the HydroSciences Montpellier laboratory at the University of Montpellier. Total alkalinity was measured by acid titration with 0.1N HCl. Major ion ($Cl^-$, $NO_3^-$, $SO_4^{2-}$, $Ca^{2+}$, $Mg^{2+}$, $Na^+$, and $K^+$) were analyzed by ion chromatography (ICS 1000 Dionex®). Precision error was $< \pm$ 5%. After acidification with 1% $HNO_3^-$, trace elements (Li, B, Sr, REE and U) were analyzed by inductively-coupled mass spectrometry using a Thermo Scientific® iCAP Q at the AETE-ISO technical platform of the OSU OREME at the University of Montpellier. The use of an in-line Argon Gas Dilution system permits the direct injection of highly mineralized samples without prior dilution. Precision error was typically <5%. Fresh water reference material SLRS-6, and seawater reference materials CASS-6 and NASS-6 for trace metals (National Research Council, Canada) were analyzed every 20 samples to monitor analytical accuracy. Mean results are





within the range of certified uncertainties. Precision error for all analyses was typically <5%.
The REE data are represented in profiles, after normalization to a reference geological material,
the North American Shale Composite (NASC) for natural waters, which corresponds to an
average sample of North American shales (Taylor and McLennan, 1985).
For stable isotopes analysis, samples collected in 2010 were analyzed at BRGM Laboratories
using a Finnigan MAT 252 mass spectrometer, whereas those collected in 2012 and 2018 were
measured on an Elementar Isoprime stable isotope mass spectrometer at the LAMA laboratory
of HydroSciences Montpellier at the University of Montpellier. Calibration was performed by
repeated analyses of in-house standards of known isotopic composition in alternation with
samples. Oxygen ($^{18}O/^{16}O$) and hydrogen (D/H) isotope ratio measurements are expressed in
parts per thousand (i.e. ‰) in the familiar δ notation relative to the Vienna Standard Mean
Ocean Water (SMOW) standard, where $\delta = ([R_{sample}/R_{standard} -1] \times 1000)$. Samples analyzed by
BRGM have a precision of ±0.8 ‰ for δD values and ±0.1 ‰ for $\delta^{18}O$ values, whereas those
determined at the University of Montpellier have an overall precision of ± 0.6 ‰ for both δD
and $\delta^{18}O$ values. Tritium analyses were performed at Hydrogeology laboratory at the Avignon
and Pays du Vaucluse University.
Sr-isotope analyses were made by thermal ionization mass spectrometry, at the BRGM
Laboratories for the 2010 samples and at the Centre de Recherches Pétrographiques et
Géochimiques in Nancy for the 2012 samples. Chemical separation of Sr was done using a Sr-
Spec ion-exchange column that has a total blank <0.5 ng for the entire chemical separation
procedure. Around 150 ng of purified Sr was loaded onto a tungsten filament and analyzed with
an average internal precision of $\pm 10.10^{-6}$ (2σ) using a Finnigan MAT262 multiple collector
thermal ionization mass spectrometer. Measured $^{87}Sr/^{86}Sr$ ratios were normalized to a $^{86}Sr/^{88}Sr$
ratio of 0.1194. The reproducibility of $^{87}Sr/^{86}Sr$ ratio measurements was tested through replicate
analyses of the NBS987 standard (0.710240) for which the mean value was $0.710232 \pm 22 \times 10^{-6}$
at the BRGM and $0.710262 \pm 13 \times 10^{-6}$ at the CRPG.
Boron isotopic ratios were measured with a Neptune+, Thermo Electron inductively-coupled
mass spectrometer (at the BRGM laboratories for samples collected in 2010 (Guerrot et al.,
2011) and at the AETE-ISO technical platform of the OSU OREME at University of
Montpellier for the other samples. The average value determined for the NIST SRM 951
standard was $4.67 \pm 0.0033$.



## 5. Results

### 5.1 Revised Conceptual Model for the Thau hydrosystem

The conceptual hydrogeological model of the Thau system at the local scale of the Balaruc-les-Bains peninsula, is summarized in Figure 2. Thermal waters rise from the deep reservoir along inferred faults in the vicinity of the Balaruc peninsula to reach the top of the Jurassic aquifer. Consistent with the recent data compilation and results acquired in the Balaruc-les-Bains area by the Dem'eaux Thau project (Ladouche et al., 2019), we propose that the Thau Thrust is likely to provide the main pathway for thermal water. The karst freshwater is found below this thermal lens as these cold waters are more dense. The diffuse intrusion of marine waters from the Mediterranean Sea into the Jurassic limestone corresponds to a saltwater wedge and is located below the karst waters which have a lower density. This saltwater wedge can nevertheless have a more complex geometry than that represented in Fig. 2, since this heterogeneous karst system is made up of multiple compartments. These contrasts in water density generate several hydraulic interfaces that change according to the different hydraulic heads resulting from continually- varying lagoon water levels, groundwater withdrawals and recharge from precipitation.

| Linear depth (m) | EC (mS cm$^{-1}$) | Temperature (°C) |
|---|---|---|
| 85 | 16.5 | 25 |
| 145 | 1 | 23 |
| 223 | 1 | 37 |
| 237 | 1 | 37 |
| 249 | 2 | 37 |
| 297 | 3.2 | 37 |
| 389 | 35 | 37 |

**Table 2 Water temperature and electrical conductivity from the exploratory drilling F13 at different depths. This inclined well (40°NW, N320°) is located near the F14 well (Fig.1b)**

This hydrogeological setting illustrated in Figure 2 was confirmed by the F13 exploratory well (Table 2), that displayed contrasting electrical conductivity and temperature over depth.

Under normal hydrologic conditions (Fig. 2a), groundwater from the karst aquifer discharges into the Vise spring, creating a plume of freshwater in the lagoon. During an occasional episode



of saltwater intrusion (or "inversac" event), the natural flow is reversed and saltwater from the
Thau lagoon flows into the Vise spring and enters the karstic aquifer via the Miocene cover
(Fig. 2b). The equilibrium between the different water bodies inside the Jurassic reservoir is
then strongly disrupted and the outflow of the karstic system through the Vise spring is
prevented. The characteristics of the thermal and karstic wells and spring in the Balaruc-les-
Bains peninsula are shown in Table 3

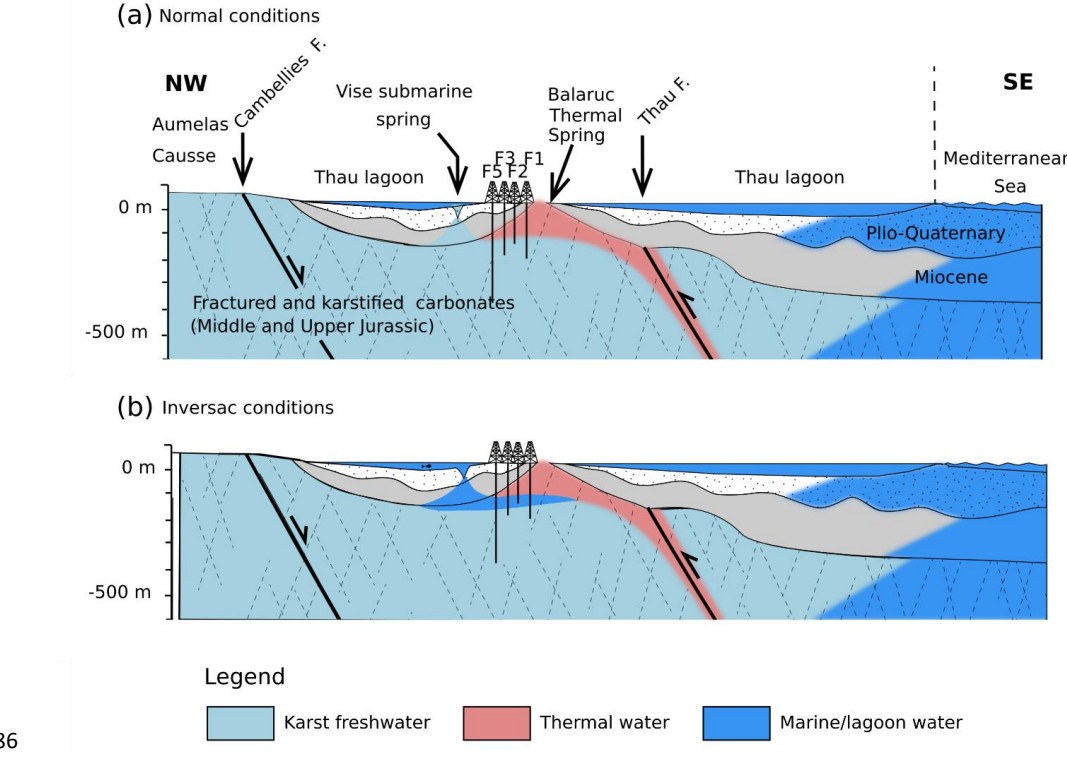


*Figure 2: Conceptual cross-section oriented NW-SE passing through thermal wells of the Balaruc peninsula, and*
*showing groundwater flows of the karst hydrosystem under (a) normal conditions and (b) inversac conditions.*
*Location of the cross-section is indicated in Figure 1a.*









| Name and BRGM location reference number | Main water type | Well depth (m) | Depth to the top of the karst aquifer (m) | Distance from the Vise submarine spring (m) |
|---|---|---|---|---|
| F5 (10165X0185) | Thermal | 105 | -72 | 325 |
| F6 (10165X0251) | Karst/thermal | 63.5 | -58.5 | 330 |
| F9 (10165X0252) | Thermal | 120 | -88 | 375 |
| F3 (10165X0183) | Thermal | 175 | -65 | 380 |
| F4 (10165X0184) | Thermal | 55 | -38 | 460 |
| F14 (10165X0257) | Thermal | 300 | -170 | 585 |
| F8 (10165X0234) | Thermal | 407 | -136 | 930 |
| Cauvy spring(10165X0021) | Karst | 0 | 0 (outcrop) | 1000 |
| CGE Tennis (10166X0212) | Karst | 115 | -95 | 2600 |
| P4 La Balme (10166X0253) | Karst | 100 | 0 (outcrop) | 2800 |
| Frescaly (10162X0194) | Karst | 54 | -22 | 3400 |
| F7-Issanka (10162X0184) | Karst | 58 | -12 | 4000 |
| F5-Issanka (10162X0136) | Karst | 27 | -14 | 4450 |
| El Cantou (10162X0197) | Karst | | | 4535 |


**Table 3 Characteristics of the main wells and springs in the study area**


**5.2 Hydraulic impact of the inversac event**

This section provides a detailed hydrodynamic analysis of the most recent inversac event in 2014 and then makes a comparison between the last three inversac events of 2008, 2010, and 2014.

The hydraulic impact of the inversac event of 2014 was observed at all Balaruc peninsula springs and observation wells (Fig.3). From the first hours after the intrusion of lagoon water, a sharp increase in water level is observed for both karst and thermal wells, although the impact is variable for different locations. The hydraulic impact was most significant for thermal well





F9, for which a +2.2 m increase in hydraulic head was observed during the first four days of the inversac event. The magnitude of the hydraulic impact is similar for F5 and F6 wells, with experienced respective increases in hydraulic head of +1.8 m and +1.9 m respectively. A lower increase in hydraulic head of just +0.7 m during the first four days of the inversac event was recorded at the Cauvy spring, situated only 1 km from the Vise submarine spring. For this spring, the increase is also partly caused by the interruption of water pumping. The hydraulic disturbance gradually increases over a 3-months period for the CGE Tennis and P4 La Balme wells, reaching a maximum of 2 m at CGE and around 1 m at P4 La Balme piezometers after 1,500 h.

Two large rainfall events occurred in September and October 2014, some 3,400 h after the beginning of the inversac event. These resulted in recharge to the aquifer that caused a sharp increase in well water levels of +1.3 m at CGE Tennis and +1 m, at P4 La Balme.

The end of the inversac event occurred in October during the normal recession period of the groundwater levels across the region. The thermal wells experienced a rapid decrease of hydraulic head, which contrasted with the slow decrease of water levels in the CGE Tennis, P4 La Balme and F7-Issanka wells. After the inversac event, the water levels in CGE Tennis, P4 La Balme and Cauvy spring were observed in an intermediate position between the maximum inversac and reference levels. In addition, the analysis of the water levels in the Thau lagoon by Ladouche and Lamotte (2015) indicates that the end of the 2014 inversac resulted from strong Tramontane winds caused an abrupt decrease in the lagoon water level in the vicinity of the Vise submarine spring. Thus, both a temporary condition of a low water level in the Thau lagoon and high waters conditions in the karst aquifer following heavy rainfall events appear to have contributed to the conclusion of the 2014 inversac event.

In the Villeveyrac Basin area, the evolution of piezometric levels during 2014 indicates that the hydrology of this region was not affected by the inversac event (Ladouche and Lamotte, 2015). The elevated piezometric level of some 70 m here compared to the Vène area (3 m) strongly suggests a compartmentalization of the karst aquifer.



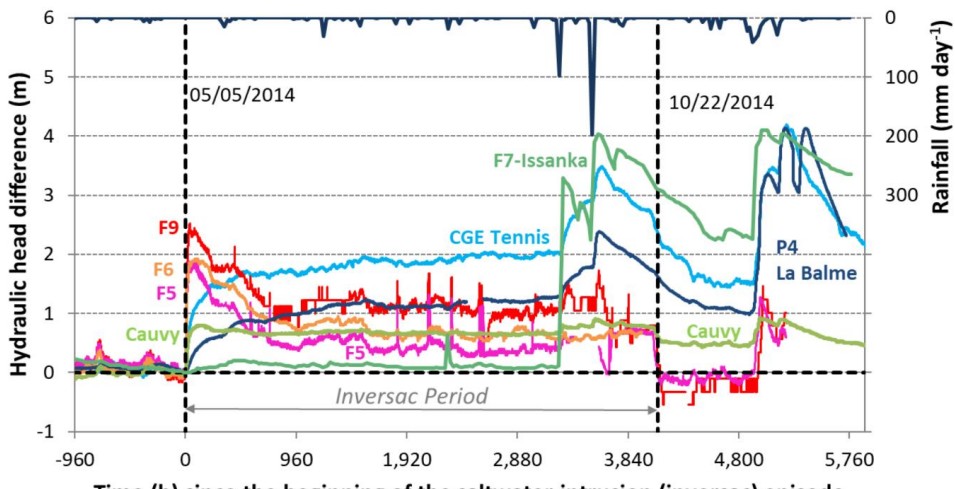


*Figure 3 Evolution of variations in hydraulic head (m) of wells and springs across the study area during the 2014*
*inversac event. The rainfall record during this time is shown at the top of the figure.*



Figure 4 illustrates the dynamics of the hydraulic disturbance to the karst system caused by the
2014 inversac event across the Balaruc-les-Bains peninsula. This detailed analysis demonstrates
the hydrologic connection between the Vise spring and the Issanka area. This perturbation
propagates rapidly and reaches a distance of 3 km upgradient in less than 24 h (i.e. a velocity
of 125 m h$^{-1}$). More specifically, the inversac causes an increase in hydraulic head of +1.5 m
throughout the peninsula and +0.5 m to the north of the peninsula (Frescaly, CGE Tennis) in
the first 24 h. This hydraulic perturbation then reaches the Issanka area in less than 72 h (i.e. a
velocity of 69m h$^{-1}$) and causes a +0.4 m increase in hydraulic head within 10 days.
In addition, Fig.4 clearly shows that the hydraulic perturbation follows a preferential pathway
with a NNE orientation. A second minor flowpath follows a ENE orientation from the Vise
spring towards Cauvy and Ambressac springs.





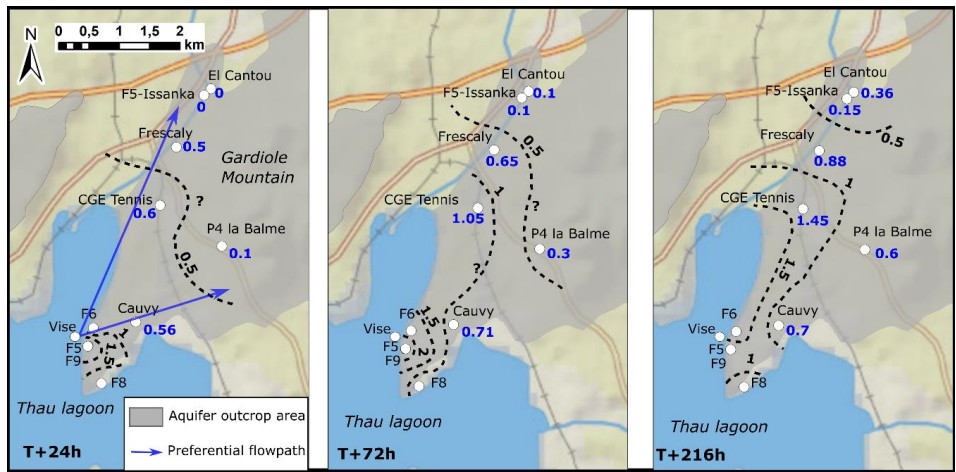


*Figure 4 Spatial propagation of the hydraulic perturbation (hydraulic head increase in m) across the Balaruc-les-Bains peninsula during the 2014 inversac event at three times after initiation: T+24 h, T+72 h, and T+216 h.*


**Comparison between the 2008, 2010, and 2014 inversac events.**

Comparison between the last three inversac events of 2008, 2010, and 2014 provides a basis for examining the hydraulic response to the phenomenon under different hydrological states. The 2008 inversac caused an increase in the hydraulic heads of all the wells and piezometers in the study area. For example, karst wells P4 La Balme and CGE Tennis showed respective increases in the hydraulic head of +2 m and +1 m.

The increase in the water levels reached the same wells during the 2014 inversac as in 2008, but was generally lower (more than 1 m for P4 La Balme and Frescaly and about 0.4 m in the Issanka area).

In contrast to the 2008 and 2014 events, the hydraulic impact of the 2010 inversac did not reach the Issanka area. An increase in hydraulic heads ranging from 1.5 to 2 m was only observed north of the Balaruc-les-Bains peninsula for the P4 La Balme and CGE Tennis wells. The explanation for this observation is discussed in the Discussion section.



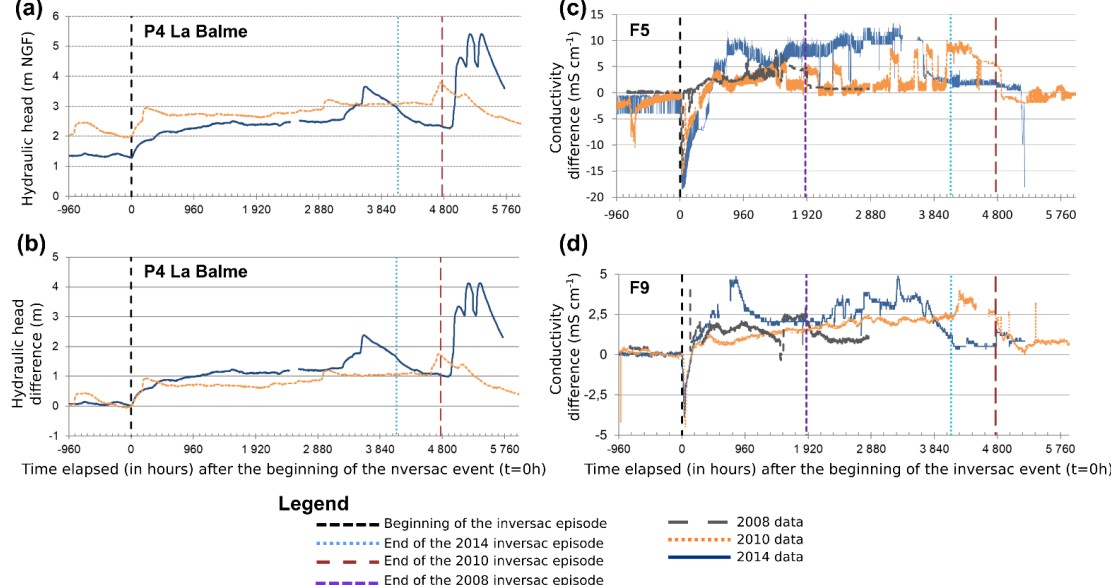

368

*Figure 5 (a) Hydraulic head (m) and (b) hydraulic head variation (m) of the karst well P4 La Balme during the 2010 and 2014 inversac events; electrical conductivity variation (mS cm$^{-1}$) for the thermal wells (c) F5 and (d) F9 during the 2008, 2010 and 2014 inversac events.*

The hydraulic head of the karst well P4 La Balme was higher during the 2010 inversac (intermediate waters) than during the 2014 inversac (very low waters) reflecting contrasted hydrologic conditions during these periods (Fig.5 a). However, the hydraulic head difference, with respect to the reference measured before the inversac (Fig. 5b), is higher in 2014 than in 2010. This suggests that the low water conditions magnify the hydraulic impact of inversac events. Similarly, electrical conductivity differences for the thermal wells F5 (Fig. 5c) and F9 (Fig. 5d) show a stronger response to the 2014 inversac than during both the 2008 and 2010 inversacs.

**5.3 Geochemical impact of the inversac event**

In addition to the hydraulic impact just described, inversac events also have a notable geochemical impact on the water bodies circulating within the karst system. As indicated in the discussion of the "Hydrogeological conceptual model" section, the salty and highly mineralized water of the lagoon mixes with thermal and karst waters to alter their quality. A multi-tracer approach was undertaken to determine the long-term hydrochemical impact of inversac events on the system that employed analysis of major ion contents, REE concentrations, and H-, O-, Sr- and B-isotopic compositions. The geochemical analyses reflect contrasting conditions: the



1996-2000 reference interval, the 2010 inversac event, and the 2012 and 2018 post-inversac
period.

**Groundwater characteristics- Major ions and water types**

Outside of the Balaruc Peninsula, karst groundwater from Issanka spring and Pézenas deep
borehole (738 m) is of a Ca-Mg-HCO3 type, characterized by electrical conductivity (EC)
values of about 500 μS cm$^{-1}$, in agreement with their karstic water type. The water from the
deep borehole has a higher temperature (37.2 °C) than that of the Issanka spring (about 17.5
°C).
Within the Balaruc peninsula, karst water from the Cauvy spring is of the same water type, but
displays elevated Cl contents and EC values of up to 222 mg L$^{-1}$ and 1200 μS cm$^{-1}$, respectively.
In contrast, the thermal waters with temperatures that range up to 49.9°C are of a Na-Cl type
with Cl concentrations of up to 7,900 mg L$^{-1}$. The Ambressac spring, located east of the Balaruc
peninsula, is of mixed water type, with higher Cl concentration than at the Cauvy spring (up to
930 mg L$^{-1}$). Some samples also have high sulfate concentrations that approach 585 mg L$^{-1}$.
The Cauvy spring and the thermal wells F5, F6, and F9 were the most impacted by the 2010
inversac event, displaying a sharp increase of conductivity as well as high contents of Cl and
Na. For example, Cl concentrations in the Cauvy spring and F6 reached levels 8 to 11 times
above values typical of normal flow conditions (i.e. 1,240 mg L$^{-1}$ compared with 157 mg L$^{-1}$ for
Cauvy and 8611 mg L$^{-1}$ compare with 759 mg L$^{-1}$ for F6). For the Cauvy spring, which is used
for potable water, this Cl content observed during the inversac event far exceeds the drinking
water standard of 250 mg L$^{-1}$.
In a post-inversac context, Cl and Na concentrations observed for the Cauvy spring and F5, F6
and F9 wells in 2012 decreased significantly, with waters at these sampling points having
returned to levels close to those of the 1996-2000 reference period. By contrast, Cl
concentrations in wells F8 and F3 increased between 2010 and 2012, suggesting an increase in
the relative contribution of the marine water component.

**Karst water and saltwater mixing traced by water stable isotopes**

As shown in Figure 6a, different waters present in the Balaruc peninsula hydrosystem are
characterized by distinct δD and δ$^{18}$O values. Normal waters within the karst system lie close
to the Local Meteoric Water Line (δD = 8 δ$^{18}$O+14; Ladouche et al., 1998), illustrating their



meteoric origin. Waters sampled from thermal wells are more enriched in D and $^{18}$O than the karst waters and are distributed along a karst-saltwater (i.e. lagoon or seawater) mixing line.

This result indicates that thermal waters result from a mixing between karst and marine end-members. However, stable isotopes information alone does not allow a determination of the origin of the marine end-member, which could be either lagoon water or modern or ancient seawater. Samples from the F8 thermal well show a slightly enriched stable isotopes signature compared to the other thermal wells, reflecting a greater contribution of the saltwater end-member in this area, as initially indicated by Aquilina et al. (2002).

F6 samples from the reference period are located within the karst water domain, confirming the strong influence of karst waters in this well (Aquilina et al. 2002) under normal flow conditions.

H- and O-isotopic signatures of groundwater from F5, F9, F8 and F6 thermal wells are shifted towards the saltwater end-member during the 2010 inversac. Then, in 2012 and 2018, their δD and $\delta^{18}$O values decreased, but still remain in an intermediate position between the inversac maximum and the reference (1996-2000) period minimum. This result raises the question of the persistence of the chemical impact in the aquifer system following occasional saltwater intrusions.

Thermal wells F14, F5 and the thermal spring S12 show similar mixing proportions for the 2012 and 2018 post-inversac times. For F5 and S12, these mixing proportions are intermediate between that of the 2010 inversac event and the 1996-2000 reference time. Groundwater samples of F3 well do not follow the same evolution, as the shift in its $\delta^{18}$O value does not appear inversac related (Fig. 6b).

441

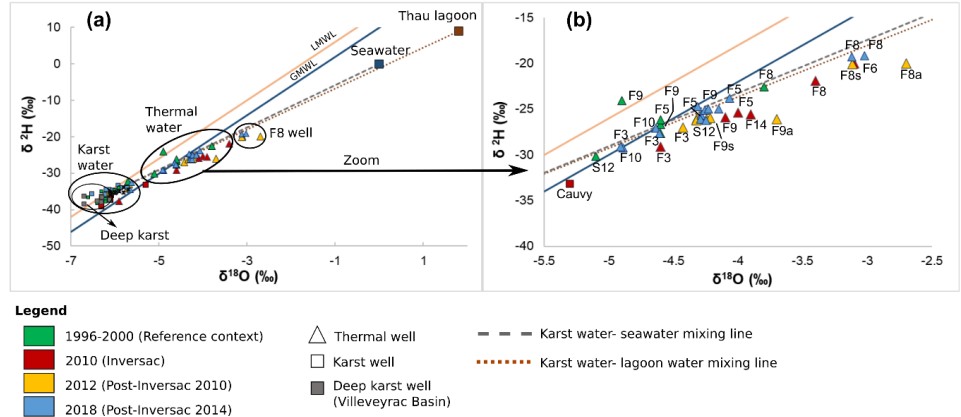

*Figure 6 H- and O-isotopic compositions, expressed as δD and δ¹⁸O values, of (a) karst waters, thermal waters,*
*F8 well water, seawater, and Thau lagoon water, showing the Global Meteoric Water Line (GMWL) after Craig*
*(1961) and the Local Meteoric Water Line (Ladouche et al., 1998 ) for reference; (b) expanded view of the thermal*
*well water array from (a) during reference, inversac, and post-inversac conditions*

Two end-member mixing calculations were made to model the geochemical impact of the 2010
inversac event on waters at the different sampling points across the study area, the first based
on stable isotope ratios and the second using Cl concentrations measured in 2010, 2012, and
2018. The proportions of karst water ($f_{fresh}$) and saltwater ($f_{salt}$) in a groundwater sample were
calculated using the following formula:

$$f_{salt} = \frac{C_{sample} - C_{fresh}}{C_{salt} - C_{fresh}}$$ equation 1

$$f_{salt} + f_{fresh} = 1$$ equation 2

Results of the mixing calculations are included in the Supplement (Table S1). This calculation
gives an indication of the magnitude of the karst water and saltwater component in of each
water sample, but does not provide additional insight into the origin of groundwater salinity.
The calculated fractions of saltwater for the karst wells and spring are similar for the three
tracers ($\delta^{18}O$, δD and Cl). However, for thermal wells with typical high Cl contents, the
fractions obtained are distinct for the different tracers, indicating another source of Cl for these
wells. Indeed, as indicated by Aquilina et al. (2002), Cl in the thermal wells is thought to derive
from both seawater and the thermal water end-member, which is itself partly constituted of old
seawater.





## Inversac persistence revealed by REE

The 14 members of the lanthanide group of elements (from La to Lu), commonly termed rare earth elements (REE), constitute an effective tool to trace water origin, water/rock interaction and mixing processes (Johannesson et al. 1997, Tweed et al. 2006, Zhan et al. 2013, Gil et al. 2018). The REE are naturally present in natural waters at trace concentrations and their normalized abundances, represented as a distribution pattern of individual REE ordered by their atomic number, provide a means of visualizing the behaviour of the entire group of elements.

The REE profiles for seawater and groundwater from the thermal wells (S12, F8, F9, F5, F6) and seawater are shown in Figure 7 for the pre-inversac reference and three post-inversac situations (Fig. 7). With the exception of the F6 well, all patterns are generally flat and show a small negative Ce anomaly. The F6 profile exhibits a more pronounced Ce anomaly and an enrichment in heavy REE. These characteristics are those of limestone REE profiles, inherited from the seawater REE profile and confirm the strong karst influence of F6 samples.

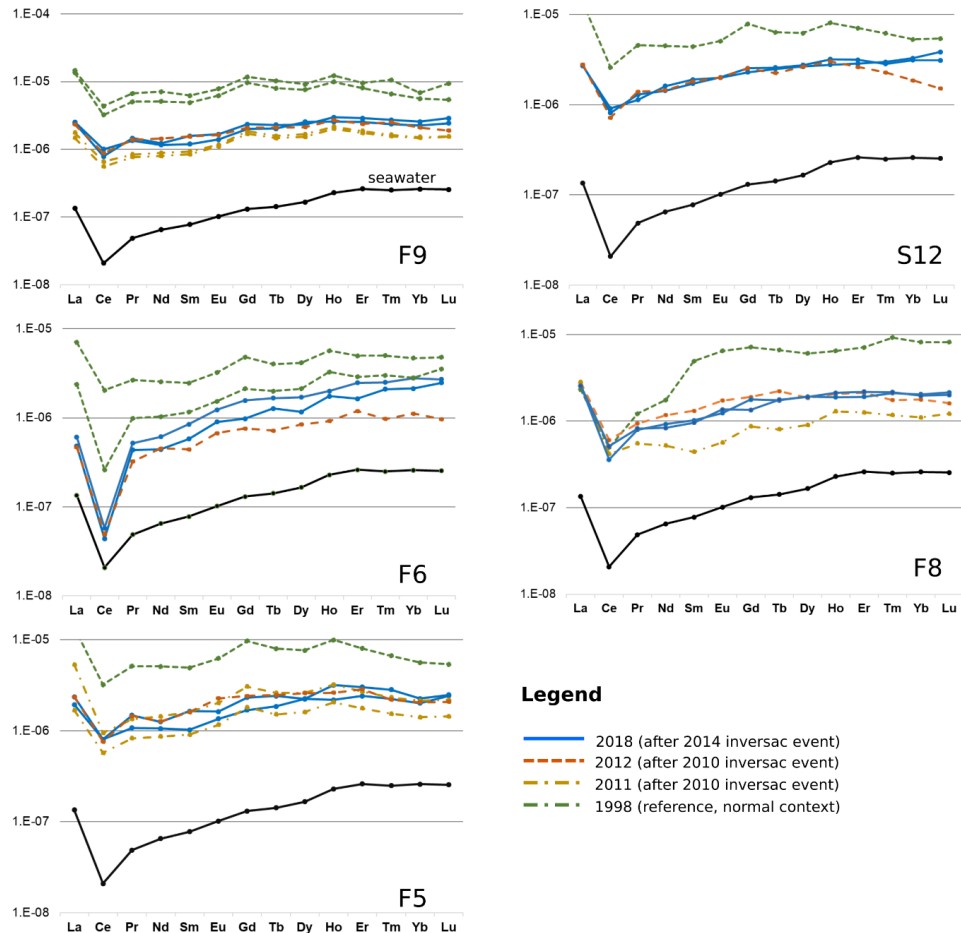

*Figure 7 REE profiles for thermal wells at different times: the 1998 data are for the pre-inversac reference state, whereas the 2011, 2012 and 2018 profiles are for the three post-inversac situations.*

REE contents during the 2011-2012 post-inversac period were generally lower than for the 1998 reference context. This is to say that, REE profiles in a post-inversac context lie in an intermediate position between the seawater profile and the reference profiles. Such an observation indicates that the thermal wells, at the time of sampling 20 months later in 2012, were still under the influence of the perturbations caused by the 2010 inversac event. The post-inversac profiles for 2012 and 2018 are quite similar, except for the F6 well, suggesting a similarity in extent of the 2010 and 2014 inversac events and post-event response of the hydrologic system. The REE profile for the F8 thermal well displays a distinctly different evolution, with an increasing shift away from the seawater profile over time. However, this well





is not hydraulically impacted by the occasional saltwater intrusion. The observed evolution does
not seem related to an increase in water withdrawals, but may instead reflect a relative decrease
in the contribution of thermal component over time.
**Characterization of the thermal and marine contributions from the strontium isotope**
**ratios $^{87}Sr/^{86}Sr$**
Consideration of strontium geochemistry allow the characterization of the thermal and marine
water contributions to the karst groundwaters. Sr-isotope ratios ($^{87}Sr/^{86}Sr$) for the groundwater
and lagoon samples are plotted as a function of strontium in Figure 8, where the 2010 and 2012
post-inversac data are compared with the 1996-2000 reference situation. It was assumed that
the waters from the F5 thermal well in the Balaruc peninsula during the reference interval are
representative of the thermal end-member, which is not affected by the karst or marine waters.
It is notable of the Sr-isotopic signature and Sr content of the CGE Tennis samples are relatively
stable in all contexts and fully representative of karst waters under confined conditions. The
Thau lagoon water sample is compositionally similar to that of the Mediterranean Sea and
corresponds to the saltwater end-member. The $^{87}Sr/^{86}Sr$ isotopic signature of karst waters and
thermal end-member are similar, indicative of derivation through water-rock interaction with
the Mesozoic carbonate aquifer. The karst end-member is different from the thermal end-
member by its lower Sr concentrations (100 µg L$^{-1}$), suggesting different water residence times.
As shown in Fig.8, all water samples plot within a ternary domain, confirming that the
geochemistry from the Balaruc peninsula groundwater can be explained by a 3-member mixing.
The karst waters from the Issanka spring that drain the Aumelas Massif have a lower $^{87}Sr/^{86}Sr$
ratios and Sr contents. The Issanka isotopic signature defines an up-gradient end-member for
the karst system, unaffected by mixings of the thermal and seawater that characterizes the
Balaruc peninsula. Waters from the P4 La Balme piezometer had a very high $^{87}Sr/^{86}Sr$ ratio in
2012, almost certainly reflecting the influence of rainfall infiltration since this sample site is
located at the outcrop of the aquifer and may be representative of surficial recharge water to the
karst aquifer.




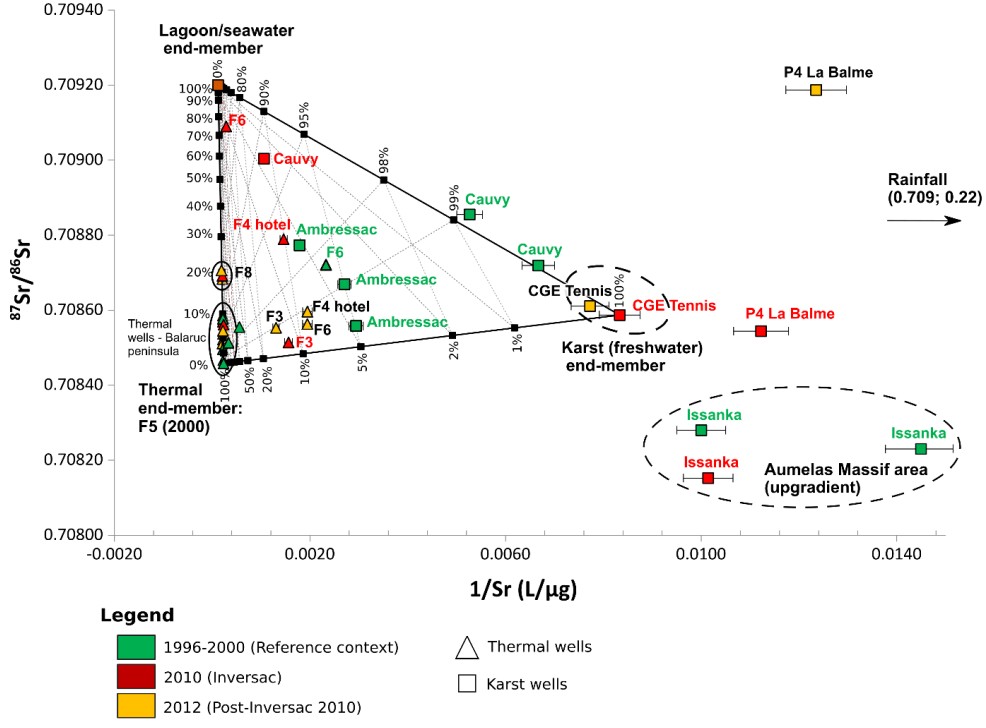


*Figure 8 Plot of $^{87}Sr/^{86}Sr$ ratio versus Sr concentration (as 1/Sr) for springs and wells of the Balaruc peninsula for different hydrogeological situations: pre-inversac reference (green), 2010 inversac (red) and 2012 post-inversac (yellow).*

The contribution of each end-member (saltwater, thermal water and karst water) to groundwater samples under different hydrogeological situations related to inversac events were estimated through 3-component mass balance equations following Faure (1986):

$$f_{salt} + f_{karst} + f_{therm} = 1 \qquad \text{equation 3}$$

where:

$(^{87}Sr/^{86}Sr)*[Sr]_{sample} = f_{salt} (^{87}Sr/^{86}Sr)*[Sr]_{salt} + f_{karst} (^{87}Sr/^{86}Sr)*[Sr]_{karst} + f_{therm}(^{87}Sr/^{86}Sr)*[Sr]_{therm}$

Results of the mixing calculations are in the Supplement (Table S2).

The Sr-isotopic composition of the Cauvy spring is quite variable during the reference interval, with samples aligned along the karst-saltwater mixing line in Figure 8. The contribution of the saltwater end-member to the spring discharge is negligible under normal flow conditions, with



the Cauvy isotopic signature very close (99%) to the karst end-member. However, the Cauvy
spring is strongly impacted by the inversac phenomenon. Although he karst end-member
remains dominant at $88.1 \pm 0.2$ %, the contribution of saltwater end-member increases to $8.1 \pm$
$0.5$ % and that of the thermal end-member reaches about $3.8 \pm 0.3$ % during the 2010 inversac
event. The contribution of thermal waters to the Cauvy spring is not surprising, as it is located
close to the old thermal spring (S12), but the large contribution of the saltwater end-member
during inversac events implies a preferential groundwater flow axis between the spring and the
Vise submarine spring. Pinault et al. (2004) have shown that variations in the contribution of
thermal waters may be related to the existence of a piston effect on the thermal reservoir,
particularly following an intense recharge event.
Thermal well waters of the Balaruc peninsula are all aligned along the thermal-saltwater end-
member mixing line in Figure 8, except for F3, F4 hotel, and F6 waters, each of which
documents a significant contribution from the karst end-member ($87.9 \pm 0.3$ %, $89.8 \pm 0.2$ %
and $53.7 \pm 0.6$ %, respectively).
The evolution of the isotopic signature for the F5, F9 and F14 borehole waters shows a similar
pattern. Thermal water $^{87}Sr/^{86}Sr$ ratios increase during the 2010 inversac event, but then
decreases in 2012 to values intermediate between the inversac condition and normal state. These
observations suggest that waters from this group of thermal wells were strongly affected by the
2010 inversac had not returned to their 1996-2000 pre-inversac Sr-isotope signature by the 2012
post-inversac situation.
The contribution of the Thau Lagoon end-member increased to $8.3 \pm 1.3$ % for the F9 waters
during the 2010 inversac, the thermal end-member was $91.3 \pm 0.6$ %, the karstic contribution
being negligible ($0.4 \pm 1.9$ %). The contribution of the Thau lagoon end-member decreased to
$5.7 \pm 1.2$ % during the 2012 context.
Reporting on water collected from the F8 borehole, Aquilina et al. (2002) noted a seawater
contribution to the thermal waters in this area of the Balaruc peninsula, which we have
estimated to be $8.8 \pm 1.3$ % for the 1996-2000 pre-inversac reference period.
The strontium concentrations and $^{87}Sr/^{86}Sr$ ratios of F8 well waters were very similar in 2010
and in April and September of 2012. Although the contribution from the saltwater end-member
during the 2010 inversac was greater than the reference condition ($18.0 \pm 1.6$ % instead of $8.8$
$\pm 1.3$ %), and remained between $17.5 \pm 1.6$ % to $21.2 \pm 1.8$ % from April to September 2012.
The thermal well F8 does not seem to be impacted by a direct intrusion of lagoon water during
the inversac event but instead influenced by seawater already within the hydrosystem.



Water at the F6 borehole was the most impacted by the 2010 inversac. During this time, the
contribution from the saltwater end-member sharply increased from $1.6 \pm 0.1$ % to $35.4 \pm 1.8$
% between 1996 to 2010 then decreased to $0.6 \pm 0.1$ % in 2012, a value below the 1996 to 2000
reference interval. Waters from the F6 borehole are representative of the upper part of the
Jurassic aquifer and demonstrate a significant contribution of the karst end-member ($94.1 \pm 0.1$
% to $91.2 \pm 0.2$ % in 1996 and 2012). This karst contribution remains significant during an
inversac context ($53.7 \pm 0.6$ %). Thus, waters from the F6 well illustrate the range of
fluctuations of the thermal component.
By contrast to the F6 well, the three end-member contributions are relatively stable for the
waters of Ambressac spring. Indeed, water samples display a large variability, with a saltwater
end-member contribution of between $1.1 \pm 0.1$ % and $2.7 \pm 0.2$ % and a karst contribution
between $94.4 \pm 0.1$ % and $91.5 \pm 0.1$ % over the reference interval (1996-2000). The thermal
contribution ranges between $4.4 \pm 0.1$ % and $5.8 \pm 0.2$ % during normal times.

**Boron Isotope Tracking of Water-rock interactions**
A lack of knowledge of the baseline B-isotopic signature during normal flow conditions does
not allow to discuss the origin of boron and its quantification for thermal borehole waters.
However, examination of $\delta^{11}B$ variations as a function of B/Cl ratio clearly identifies water
interactions with clay minerals. Indeed, boron is present in aqueous solution, in the form of
boric acid $B(OH)_3$ and borate ions $B(OH)_4^-$. The distribution between these two species depends
on pH, temperature and salinity (Hershey et al 1986, Hakihana 1977). For the boron isotopes,
$^{11}B$ is preferentially incorporated into boric acid whereas $^{10}B$ has a greater affinity for the borate
ion. During the water-rock interaction, $B(OH)_4^-$ will be preferentially adsorbed onto clay
minerals or organic matter, resulting in a decrease of water B concentration and a consequent
enrichment of $^{11}B$ in residual water.
Figure 9 shows the Cl/B ratio as a function of B isotopic composition. $\delta^{11}B$ of the Thau Lagoon
is 39.1 ‰, in agreement with the signature of the modern seawater of 39.5 ‰; (Aggarwal et al.,
2004, Aggarwal and Palmer 1995). By contrast, karst waters display a range of $\delta^{11}B$ values
from 23.1 to 52.3 ‰, but are strongly $^{10}B$ enriched relative to seawater signature, whereas,
thermal waters have $\delta^{11}B$ values of up to 44.3 ‰ that are similar to or higher than the seawater
signature.



There is a good alignment of the data points in Fig. 9, which illustrates the process of water/rock
interaction and indicates that the B in the waters across the Balaruc peninsula is derived from
clay sediments. That 11B/10B ratios are well above the seawater values, suggests that water
from the Thau lagoon interacts with clay sediments within the karst system during the
occasional intrusion into the carbonate aquifer. An exploration dive of the bottom of the Vise
spring in April 2017 showed the absence of sediment on the walls and bottom of the spring.
The inversac phenomenon therefore seems to be accompanied by a significant intrusion of
sediments (clayey marine mud, rich in organic matter). This phenomenon seems to contribute
to initiate the process of boron isotope fractionation. This process could also occur within the
silty-clay Miocene unit during the transfer of water from the lagoon to the Jurassic aquifer. This
Miocene formation overlies the Jurassic aquifer and is about 30 m thick at the Vise spring. The
high isotopic enrichment observed for the F3 borehole waters can be explained by their high
pH values (8.3) and thus a relatively higher proportion of the $B(OH)_4^-$ species, leading to a
higher adsorption on the solid phase and a higher enrichment of $\delta^{11}B$ of the residual water.
Boron isotopes are used to identify the chemical modification of the water that enters the system
and interacts with the clay matrix. There is therefore a modification of the water chemistry
before it even enters the karst aquifer, and this modification is initiated during the saltwater
intrusion.

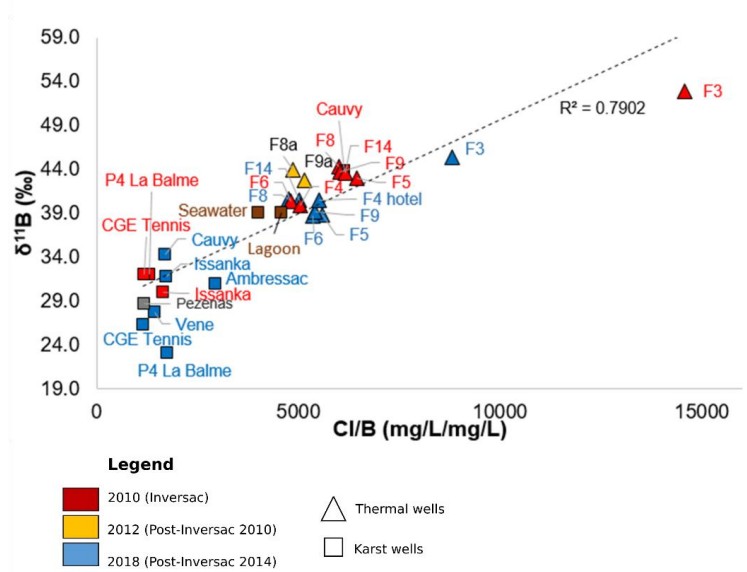






*Figure 9. Plot of δ¹¹B versus Cl/B ratio for different hydrogeological contexts: 2010 (inversac times in red),*
*2012 and 2018 (post-inversac times in yellow and blue)*

## 6. Discussion

### 6.1 Implications for Dynamics of the Thau hydrosystem

**Extent of the hydraulic impact and conditions required to reach the upgradient area of Issanka**

Analysis of the hydraulic response of the Thau hydrosystem to the last inversac events provides insights into the dynamics of the system and a better understanding of the underlying controlling mechanisms. For example, the 2010 inversac did not impact the upgradient area of Issanka in contrast to the 2008 and 2014 inversac events. This is attributed to the fact that the karst system in 2010 was less in deficit than at the time of the 2008 and 2014 events, when very low flow conditions prevailed. For example, piezometric levels measured for the Issanka area in 2008 and 2014 were lower than in 2010 (i.e. 6.82 m in 2008 and, 6.89 m in 2014 and, 9.33 m in 2010). Considering that the overflow level of the Issanka spring is at ~9.5 m, this implies that the hydraulic impact in the Issanka compartment is controlled by the pressure head difference between the upstream (Issanka) and downstream (Balaruc peninsula) compartments. Piezometric monitoring at these different times illustrates that the occurrence of the inversac phenomenon is mainly determined by hydrogeological processes that occur within the downstream compartment of the karst system. Water level of the Thau lagoon is one of the controlling parameters. At inversac initiation in 2008, the water level in the Thau lagoon was higher than in 2010 at 1 m compared to just 0.5 m, but then only of 0.2 m in 2014. For the inversacs of 2008 and 2010, the sudden rise in lagoon waters of + 0.5 m and + 0.4 m, respectively, in a few hours is considered to be
the triggering factor. By contrast, no sudden change in the water level of lagoon was observed in 2014. The triggering mechanism of this event is still poorly understood, however it is likely that the withdrawals from the aquifer for drinking water and irrigation purposes as well as the low recharge during the previous winter may be the explanatory factors.

Thus, two factors would need to be combined for an inversac to reach the upgradient area of Issanka: (i) first, low water conditions that produces a very low hydraulic head throughout the aquifer. The hydraulic head would need to be lower than 7 m in the Issanka area, i.e. below than the hydraulic head imposed by the saltwater intrusion; (ii) an event-specific triggering



mechanism, such as a sudden rise in the water level of the Thau lagoon or groundwater withdrawals.

The intrusion of lagoon waters in the system causes a 2-phase response within the Thau hydrosystem:

1) A short initial transient phase of a few days duration during which there is a reorganization or displacement of the different karst, thermal and marine water bodies within the hydrosystem,

2) A longer transient phase during which the hydrosystem reaches a new equilibrium (in 30 days in 2008 and 2010, 40 days in 2014) with clear evidence of water mixing causing important changes in the physico-chemical parameters of waters from the sampling points F5, F6, F9 and the Cauvy spring.

**The impact on water quality is local whereas the hydraulic impact is perceived at the regional scale**

Results in section 5.1 showed that the hydraulic impact of an inversac event is instantly propagated throughout the Balaruc peninsula and the effects of such events can also reach longer distances into the karst system than previously described. The return to the equilibrium situation occurs within days after a significant recharge event and is much more rapid than the dissipation of the chemical impact of the inversac event. Indeed, the 2010 and 2014 inversac events lasted about 6 months and had a persistent chemical impact on the hydrosystem. It stayed perceptible on the quality of the water in certain compartments from 20 months to 42 months after the occasional inversac intrusions ceased.

Even though the geochemical conditions within the hydrosystem were not monitored during the 2014 inversac event, the 2018 data are similar to those for 2012, suggesting that the hydrosystem did not reach its geochemical reference conditions and was still under a state of influence by the latest inversac event.

Geochemical analysis indicates that the 2010 inversac had the greatest impact on waters from the F6 well and the Cauvy spring, thus confirming the existence of a preferential groundwater flowpath between the Vise spring and those points.

The physico-chemical disturbances resulting from inversac event are not observed at sampling points CGE Tennis, P4 La Balme, Issanka located in the upgradient area of the Balaruc





peninsula. The geochemical impact is, therefore, restricted to within a 1-kilometer radius from the Vise spring (Fig. 10).

This slow geochemical recovery is best explained by the characteristic internal heterogeneity of the karst aquifer, which is typically described in terms of a triple porosity model comprising: (i) matrix or primary porosity, (ii) fracture porosity and (iii) conduit porosity (Palmer et al 1999; Martin and Screaton 2001). Influx of inversac water into the matrix porosity of the karst aquifer, its storage, and then prolonged time of removal and return to fracture and conduit flow explains the persistent fingerprint of the saltwater intrusions in the Balaruc peninsula.

Wells F8 and F10 in the southern part of the Balaruc peninsula were not affected by the inversac events. This part of the hydrosystem corresponds to a deeper compartment in the karst system, where the top of the Jurassic aquifer is 136 m deep in the F8 well and 195 m deep in the F10 well, which appears to be less hydraulically connected to the rest of the peninsula. Previous studies postulated that an indirect impact of an inversac event might occur (Ladouche et al., 2012). However, the increasing mineralization of F8 waters over time from 20 mS cm$^{-1}$ in 1996 to 26.5 mS cm$^{-1}$ in 2018 (Figure S1 in the Supplement) suggests that waters from this well are not influenced by the occasional saltwater intrusions into the hydrosystem. This distinct evolution cannot be explained by pumping withdrawals as well withdrawals rates in the F8 thermal well were steady over the period of inversac monitoring. However, a decrease in the recharge rate from the karst system could result in an increase in the proportion of the thermal or marine end-member contributions. Recent monitoring of springs (Ambressac and S12) will address this question in the Dem'Eaux Thau project.

A comprehensive review of historical inversac events undertaken as part of this study showed that there is a hydraulic connection between the Villeveyrac Basin and the Balaruc peninsula. The relationship between these two regions is clearly demonstrated as dewatering of a bauxite mine in the Cambelliès area (Fig.1) triggered the 1967 inversac event.



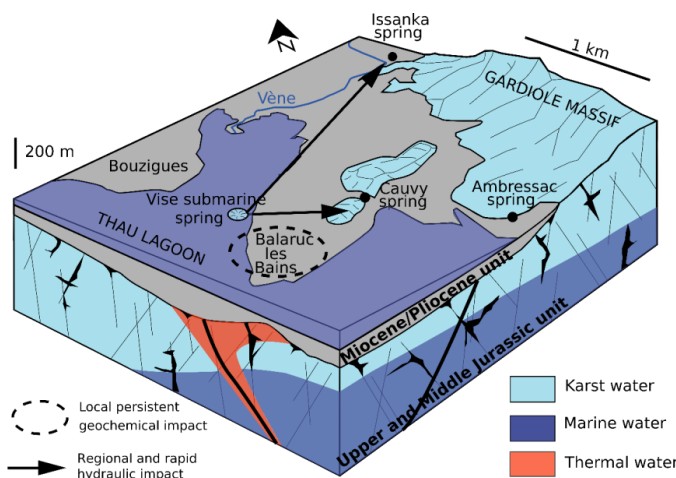

*Figure 10. Bloc diagram of the study area showing the spatial extent of the hydraulic and geochemical impacts.*
**6.2 Implications on groundwater management:**
Results presented here illustrates the fragile equilibrium that exists between different
groundwater bodies in the Thau hydrosystem. During low flow conditions, any modification of
the exploitation regime of the system can cause an inversac event and cause deleterious long-
term consequences.
Historically, the 1967 and 2010 inversac events led to the complete cessation of withdrawal
from the Cauvy spring, followed by its subsequent abandonment because the quality of this
major drinking water supply was strongly impacted by the inversac events.
The Issanka spring is another major source of drinking water in the area for a population of
40,000. This study recognized the conditions needed for the hydraulic impact of an inversac
event to reach the Issanka well-field. Such new understanding can support water management
operations by facilitating the prediction of the arrival of an inversac disturbance, using the
criteria highlighted in section 6.1.
While the southern compartment of the hydrosystem has not been impacted by inversac events,
the gradual increase in mineralization observed for the F8 thermal well has the potential to
affect its quality over time. In order to sustain the water quality of this thermal well, further
studies should investigate the specific dynamics of this karstic compartment and the underling
mechanism explaining this observation. More specifically, the hydraulic connection between
this deep compartment with the Sète thermal system south of the study area should be
investigated.
Finally, unlike classical saline intrusion, the inversac events mobilize sediments loaded with
organic matter during saltwater intrusions from the Vise spring. This intrusion of sediments
may have an additional impact on the water quality within the karstic hydrosystem as suggested
by B-isotopic signatures. Future studies could explore the fate and transport of the organic
matter in the karst system.

### 7. Conclusion

This study investigated the impact of occasional saltwater intrusions (inversac in French) into
the karst hydrosystem of the Balaruc peninsula and provides an improved understanding of the
hydrodynamics and recovery of the aquifer from saltwater intrusions. Differences were
observed between the hydraulic and geochemical responses, both spatially and temporally. The
karst, thermal and marine water bodies within different compartments of the karst system
coexist within a very delicate equilibrium that is disturbed by the rapid intrusion of lagoon
waters during inversac events. While the hydraulic impact of such saltwater intrusions is
immediate and manifest over a distance of some 5 km within the hydrosystem, the geochemical
impact focused within a 1 km radius around the Vise spring but temporally persistent.
Geochemical tracers showed that the hydrosystem did not reach its pre-inversac reference state
20 months and 40 months after the occurrence of inversac of 6-months duration. The slow
geochemical recovery of the Balaruc peninsula karst system reflects the triple-porosity
character of the karst aquifer. Additionally, preliminary results suggested that the modification
of the groundwater chemistry is initiated by an interaction process with clay sediments and
organic matter during the saltwater intrusion event and possibly during the subsequent transfer
of seawater to the karst aquifer through the Miocene formation. This study demonstrates the
potential to couple physico-chemical, hydrogeological and isotopic data to understand the
complex inversac phenomenon and recognize preferential flowpaths during such events.
The results of this study support groundwater management across the Balaruc peninsula. The
triggering factors of the saltwater intrusions were highlighted as well as the conditions needed
for the saltwater intrusion to hydraulically impact the upgradient area, which is a major source
of drinking water supply. While the water quality of some of wells across the Balaruc peninsula
are strongly impacted by the inversac and retain a geochemical signature of the event, the
thermal wells located in the southern portion of the hydrosystem are not affected by inversac
events, suggesting hydraulic isolation from the remainder of the Balaruc peninsula.



## 8. Data availability

The geochemical and isotopic data are accessible at https://doi.org/10.5281/zenodo.3893897

## 9. Supplement link

The supplement related to this article is available online at

## 10. Authors contribution

BL, JLS and CL designed the study and data collection plan. MAP, JLS and BL conducted the field work, collected the hydrogeological and geochemical data, and performed the data analysis. MAP prepared the paper with contributions from RH and JLS. All authors reviewed and edited the manuscript.

## 11. Competing interests

The authors declare that they have no conflict of interest

## 12. Acknowledgements

This work was funded by a FEDER-CPER grant (Agence de l'Eau, Balaruc-les-Bains, SMBT, and BRGM) through the Dem'Eaux Thau project. Authors are thankful to the municipalities of Balaruc les Bains and Pézenas for continuous support during the 2018 field work. Thanks are due to Franck Bujaldon (city of Pézenas), Ludovic Sarrou (Thermal spa, Balaruc les Bains), Nathalie Masscheleyn (Balaruc les Bains municipality), David Mimard (Suez) and Vincent Durand (Antéa) for granting access to the thermal boreholes, karst springs and piezometers. We are thankful to Jean-Gilbert Muller from President Electronics for providing access to the Ambressac spring. Gilles Lorente and Kevin Buttaro (Syndicat Mixte du Bassin de Thau) are thanked for their technical support during the March 2018 field campaign. We also acknowledge Nicolas Patris (HydroSciences Montpellier) for his help and useful discussion on stable isotopes. Remi Freydier (HydroSciences Montpellier) and graduate student Hikma Kassime are also thanked for their valuable contribution on the B isotopic analyses. We thank Russell Harmon (North Carolina State University) for a thorough review of an earlier version of the manuscript.

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
