# Peer review of "Hydraulic and geochemical impact of occasional saltwater intrusions through a submarine spring in a karst and thermal aquifer (Balaruc peninsula near Montpellier, France)"

_Hydrology and Earth System Sciences, 2020_

## Referee Comment (RC1) · Ekkehard Holzbecher (Referee) · 27 Jul 2020

The paper is an excellent study in which different methodologies (hydrogeological, physical and geochemical) and data are combined to obtained a coherent view of the system. I suggest few technical corrections, which I list here:

Line 94: Table 1, do not justify text within table cells

Line 272: Table 2, use centered format in all cells, including header

[Figure]

Line 294: Table 3, do not justify text within table cells

Line 346: leave blanc between number and unit, 69 m h-1

Line 368: Figure 5, the 2x2 sub-figure design extends outside of the page margins; as the reader may want to see the details, I suggest to put sub-figures a-d vertically in a 4x1 design

Line 450: use italics for ffresh and fsalt, in order to match with format in the equations

Line 528: use italics for the f-factors, in order to match with format in equation 3

---

## Referee Comment (RC2) · Anonymous Referee #2 · 8 Sep 2020

Authors used multi-tracer approach to understand occasional saltwater intrusions in a karst coastal aquifer in southern France. The topic fits with the journal scopes. In general the paper is well written. My major concern is the novelty of this work. The methodology and approach are standard. From this point of view, the paper does not present any novelty. However, the novelty relies on the study area. But this point is briefly discussed in the paper. I am suggesting minor revision. I would like to ask the authors to revise the introduction to point out the novelty of this work by discussing more previous studies on understanding occasional saltwater intrusions in the study

area and some previous works related to the methodology.

---

## Author Comment (AC1) · 11 Sep 2020

- **Response to Referee 1**

**We thank Dr. Holzbecher for his careful and useful review of our manuscript. Below we respond (in bold type) to Dr. Holzbecher's specific comments** (in normal type)

RC1: The paper is an excellent study in which different methodologies (hydrogeological,physical and geochemical) and data are combined to obtained a coherent view of thesystem. I suggest few technical corrections, which I list here:

Line 94: Table 1, do not justify text within table cells

**Yes, we will make the suggested change**

Line 272: Table 2, use centered format in all cells, including header

**Yes, we will fix this**

Line 294: Table 3, do not justify text within table cells

**Yes, we will correct that**

Line 346: leave blanc between number and unit, 69 m h-1

**Thank you for catching that, we'll fix it**

Line 368: Figure 5, the 2x2 sub-figure design extends outside of the page margins; as the reader may want to see the details, I suggest to put sub-figures a-d vertically in a 4x1 design

**We agree with the comment and we'll make the suggested changes**

Line 450: use italics for ffresh and fsalt, in order to match with format in the equations

**Thank you, we'll fix that**

Line 528: use italics for the f-factors, in order to match with format in equation 3

**Yes, we'll use italics for consistency**

- **Response to Referee 2**

**We would like to thank Referee 2 for the constructive feedback and the suggestions to improve the manuscript. Below are our responses (in bold type) to the referee's specific comments** (in normal type)

RC2: Authors used multi-tracer approach to understand occasional saltwater intrusions in a karst coastal aquifer in southern France. The topic fits with the journal scopes. In general the paper is well written.

**Thank you for this positive feedback**

My major concern is the novelty of this work. The methodology and approach are standard. From this point of view, the paper does not present any novelty. However, the novelty relies on the study area. But this point is briefly discussed in the paper. I am suggesting minor revision. I

would like to ask the authors to revise the introduction to point out the novelty of this work by discussing more previous studies on understanding occasional saltwater intrusions in the study area and some previous works related to the methodology.

**We will revise the introduction to better reflect the novelty of our work, as suggested by Referee 2.  As pointed out by Referee 2, the novelty relies in the study area. To the best of our knowledge, the literature on occasional saltwater intrusions through a submarine karst spring is very limited. Our work stands out from the previous studies in the study area in part because our multi-tracer approach includes more tracers and consider multiple inversac events since 1967. Also, previous studies in the Thau basin focused on the hydrochemistry while our work is the first to combine a multi-tracer approach with hydrogeological data to fully describe this phenomenon. Our comprehensive approach was helpful in developing a new conceptual model of the site and provide insights on the management of the groundwater resources.**

---

## Author Response (AR1)

**Final response to referees**

We would like to thank referees and the Associate Editor for their valuable comments that helped enhance the quality of the paper.

We took into account all the comments provided by the referees. All the modifications are described in the following text. Line numbers refer to those of the marked manuscript.

- **Response to Referee 1**

**We thank Dr. Holzbecher for his careful and useful review of our manuscript**. **Below we respond (in bold type) to Dr. Holzbecher's specific comments** (in normal type)

RC1: The paper is an excellent study in which different methodologies (hydrogeological,physical and geochemical) and data are combined to obtained a coherent view of thesystem. I suggest few technical corrections, which I list here:

Line 94: Table 1, do not justify text within table cells

**Agree, changes made (line 95)**

Line 272: Table 2, use centered format in all cells, including header

**Agree, changes made (line 279)**

Line 294: Table 3, do not justify text within table cells

**Agree, changes made (line 301)**

Line 346: leave blanc between number and unit, 69 m h-1

**Agree, changes made (line 353)**

Line 368: Figure 5, the 2x2 sub-figure design extends outside of the page margins; as the reader may want to see the details, I suggest to put sub-figures a-d vertically in a 4x1 design

**We agree with the comment and we made the suggested changes. Subfigures in Figure 5 are now presented vertically (line 375)**

Line 450: use italics for ffresh and fsalt, in order to match with format in the equations

**Agree, changes made (line 457)**

Line 528: use italics for the f-factors, in order to match with format in equation 3

**Agree, changes made (line 535)**

- **Response to Referee 2**

**We would like to thank Referee 2 for the constructive feedback and the suggestions to improve the manuscript. Below are our responses (in bold type) to the referee's specific comments** (in normal type)

RC2: Authors used multi-tracer approach to understand occasional saltwater intrusions in a karst coastal aquifer in southern France. The topic fits with the journal scopes. In general the paper is well written.

**Thank you for this positive feedback**

My major concern is the novelty of this work. The methodology and approach are standard. From this point of view, the paper does not present any novelty. However, the novelty relies on the study area. But this point is briefly discussed in the paper. I am suggesting minor revision. I would like to ask the authors to revise the introduction to point out the novelty of this work by discussing more previous studies on understanding occasional saltwater intrusions in the study area and some previous works related to the methodology.

**We revised the introduction to better reflect the novelty of our work, as suggested by Referee 2 (lines 48-50, 99-102 and 110-111). As pointed out by Referee 2, the novelty relies in the study area. To the best of our knowledge, the literature on occasional saltwater intrusions through a submarine karst spring is very limited. Our work stands out from the previous studies in the study area in part because our multi-tracer approach includes more tracers and consider multiple inversac events since 1967. Also, previous studies in the Thau basin focused on the hydrochemistry while our work is the first to combine a multi-tracer approach with hydrogeological data to fully describe this phenomenon. Our comprehensive approach was helpful in developing a new conceptual model of the site and provide insights on the management of the groundwater resources.**

**Other improvements:**

- **We updated the references list to comply with the HESS standards.**
- **Line 162, we changed "Paloc et Bonnet 1969" with "Bonnet and Paloc 1969" and updated the associated reference in the references list.**
- **Line 269, we changed the reference "Hemelsdael et al." with "Ladouche et al 2019".**
- **Line 332, we added the word "that" in the sentence.**
- **Line 464, we deleted the word "of".**
- **Line 481, we deleted "(Fig.7)" as Figure 7 was already cited in the sentence.**
- **Line 559, we added the word "and" in the sentence for clarity.**
- **Line 559-600; we changed "Cl/B ratio as a function of B isotopic composition" with "B isotopic composition as a function of Cl/B ratio".**
- **Line 602: we changed 52.3‰ with 43.8‰.**
- **Line 618, we deleted the word "enrichment".**
- **Line 766: we changed the word "remainder" with "rest".**
- **In the Supplement, we modified the title of Table S2 by changing "2018" with "1996-2000".**

[revised manuscript text omitted]